# Non-Invasive Wildlife Disease Surveillance Using Real Time PCR Assays: The Case of the Endangered *Galemys pyrenaicus* Populations from the Central System Mountains (Extremadura, Spain)

**DOI:** 10.3390/ani13071136

**Published:** 2023-03-23

**Authors:** Adriana Ripa, José A. Díaz-Caballero, María Jesús Palacios-González, Javier Zalba, Antonio Espinosa, Juan Luis García-Zapata, Ana Gómez-Martín, Vasyl Tkach, José Luis Fernández-Garcia

**Affiliations:** 1Genetic and Animal Breeding, Faculty of Veterinary, University of Extremadura, 10071 Cáceres, Spain; 2Dirección General Sostenibilidad, Consejería Para la Transición Ecológica y Sostenibilidad, Junta de Extremadura, 06800 Merida, Spain; 3Department of Mathematics, University of Extremadura, 06006 Badajoz, Spain; 4Department of Biology, University of North Dakota, Grand Forks, ND 58201, USA

**Keywords:** Central System Mountains, *Galemys pyrenaicus*, insectivores, non-invasive, *Omphalometra flexuosa*, pathogens, real time PCR

## Abstract

**Simple Summary:**

Until now, data on parasites or bacteria of the Iberian desman (Galemys pyrenaicus) was practically absent. We used non-invasive methods of sample collection and analysis to determine the health status of G. pyrenaicus. We detected four species of bacteria and three parasites using qPCR assays. Based on DNA sequence data, G. pyrenaicus in the study area harbors a likely new species of the previously monotypical genus *Omphalometra*.

**Abstract:**

The Iberian desman *(Galemys pyrenaicus*) is a small semi-aquatic mammal that inhabits mountainous areas from the centre to the north of the Iberian Peninsula and the Pyrenees and is listed as endangered because it has suffered a serious decline. Since 1960, only three species of digeneans (*Omphalometra flexuosa, Maritrema pyrenaica* and *Mathovius galemydis*) and two nematodes *(Aonchotheca galemydis* and *Paracuaria hispanica*) have been reported from the desman, but no further information on health status and no data from Extremadura has been available. The aim of our study was to characterise the diversity and distribution of parasites and microbiomes of desmans in different areas of the Central System of Extremadura. Between 2019 and 2021 we collected 238 fecal samples and one tissue (intestine) sample that was obtained from a dead desman. DNA templates were processed by commercial or customised real-time PCR using TaqMan probes. Representative data were obtained for *Cryptosporidium* spp., *Omphalometra* spp., *Eimeria* spp., *Salmonella* spp., *Staphylococcus* spp. and *Leptospira* spp. *Omphalometra* spp. was studied using a newly developed PCR test. The screening of the dead desman allowed us to obtain, for the first time, a partial sequence of the 18SrDNA. This study is the most complete study of the desman, allowing us to identify parasites and the microbiome in populations of *G. pyrenaicus* using non-invasive sampling.

## 1. Introduction

The Iberian desman, *Galemys pyrenaicus* (E. Geoffroy Saint-Hilaire, 1811), is a small semi-aquatic mammal, endemic to mountainous areas from the middle to the north of the Iberian Peninsula (northern and central Spain, northern Portugal), and the Pyrenees (France, Spain and Andorra) [1]. The species is listed as Endangered by the IUCN [2] and considered to be of high conservation concern [3] because desman populations have suffered a substantial decline in the last 20 years across its entire range [4,5]. This species has highly specific environmental and ecological requirements, which have been suggested among the main causes of the significant decline in some of its populations [6]. Several studies have reported a high rate of extinction of its population in the French Pyrenees [7]. Compared to most other mammals, studies on the desman are scarce; in part, due to the difficulty of its capture and handling, as well as existing restrictions. In addition, the distribution of this species in Extremadura is limited to small populations [1], which impairs their sampling and study.

This is especially important considering that ongoing climate change may promote the expansion of host ranges, while increased human–animal interactions enhance the risk of spillovers [8,9]. For several decades, members of orders Rodentia and Eulipotyphla have been recognized as reservoirs of pathogens, some of them causing zoonotic diseases in humans [10], which can have a dramatic impact on the economy and public health [8]. The ability of rodents and shrews to carry a wide range of microbial pathogens has been well-documented [10]. These small mammals may serve as definitive and intermediate hosts of ectoparasites and amplify reservoirs for directly transmitted diseases [11]. Endangered species usually have small population sizes and small and/or fragmented distribution ranges, which leads to the elevated risk of extinction due to a variety of potential causes, including pathogens [12,13,14].

It is worth noting that gregarious populations of small mammals are more likely to carry pathogens than solitary ones [15], as in the case of desmans, due to the limited vagility of the latter. Timon David (1960) [16] was the first to report a parasite from the Iberian desman. It was a digenean *Omphalometra flexuosa*, which was also the most frequently reported parasite in subsequent studies. This parasite was found in desman populations from the French Pyrenees as a variation of *Omphalometra flexuosa var. peyrei*. Later, Vaucher (1975) [17] reported its presence in the southernmost desman population in the Iberian peninsula, which inhabits the Central System Mountains in the province of Salamanca. Other parasitological studies reported two additional digenean species, *Maritrema pyrenaica* [18] in Spain and France [19,20,21] and *Mathovius galemydis* [22] in Spain. Besides digeneans, two species of nematodes, *Aonchotheca galemydis* [23] and *Paracuaria hispanica* [24], were described from the desman in northern and central Spain. To the best of our knowledge, there is no published information on microbial symbionts/pathogens of *G. pyrenaicus.* Due to the state (degradation due to exposure to elements) and size (droppings are less than 200 milligrams in most cases) of available samples, we focused on the groups of parasites known in desmans (such as digeneans), or important pathogens reported from other small mammals, for which we could develop our own assays or obtain commercially available diagnostic kits. Based on previously available knowledge from insectivores or other small mammals, the pathogens covered in our study, are among those commonly affecting animal health. In this study, we attempted to identify a wide range of bacteria, protozoans and helminths in the Iberian desman using non-invasive molecular techniques.

## 2. Materials and Methods

### 2.1. Study Area and Sampling

The study was conducted in the autonomous community of Extremadura (Spain) in the western part of the Iberian Peninsula. All samples were collected from 2018 to 2021.

The samples were collected as a part of the monitoring project “Gestion Integrada de la Biodiversidad en el área Transfronteriza” (BioTrans) and geotagged, but by the decision of the regional government, the availability of the geolocation data is restricted to avoid public knowledge due to the worrying state of conservation of the species.

The sampling sites were determined according to the Recovery Plan for the Desman (*Galemys pyrenaicus*) in the Central System (Iberian Peninsula) in Extremadura (DOE 158 08/14/2018) [25]. According to this plan, the study was carried out in three administrative regions: Ambroz valley (shortened: Ambroz), Jerte valley (shortened: Jerte) and La Vera (Table 1). The zoning was established in riverbeds and riverbanks in the areas of the known distribution of the species in Extremadura, as well as other areas with adequate habitats as follows: (1) critical areas (n = 10) defined as sites with current presence through genetic validation records, taking the cut-off point when the area downstream failed to meet the appropriate habitat conditions; (2) areas of importance (n = 1) of transitory presence but playing a key role for gene flow and; (3) favourable areas (n = 13) as those currently having a quality habitat and evidence of historical or recent presence (last 30 years). In order to comply with the commitment made that the data should be confidential, the sampling sites within an area (whether critical, important or favourable) were named “Sampling Sections” and randomly coded with a number between 1 and 24.

We collected a total of 238 samples of faeces that were macroscopically compatible with those deposited by *G. pyrenaicus* along the riverbeds and one tissue sample (intestine) from a male desman that was found dead (La Vera population) (Figure 1). The samples were transported in tubes containing ethanol (96% or higher) and stored at −20 °C during collection in the field and at −70 °C upon transfer to the laboratory.

The distribution data were analysed by the criteria of sampling years and geographic distribution based on localities (administrative district, zoning, and sampling sections (by river) (Table 1).

### 2.2. DNA Extraction

DNA from faecal samples was isolated using the QIAamp^®^ Fast DNA Stool Mini Kit (QIAGEN GmbH, Hilden, Germany) following the manufacturer’s instructions. However, DNA from tissues was extracted using a fast salting out procedure [26]. The DNA extractions were quantified using a Qubit 4 Fluorometer (Thermo Fisher Scientific, Waltham, MA, USA) and stored at −70 °C until use.

### 2.3. Sample Genotyping 

Prior to the screening for microbes and parasites, all faecal samples were genotyped to ensure that they belonged to the desman. This was performed by two alternative methods: initially, the RFLP-PCR was used as described by Fernández-García and Vivas-Cedillo [1]; later, it was substituted by a duplex probe-based real-time PCR assay (Ripa et al., in prep.) that allow discrimination between desman and Mediterranean water shrew faeces. Real-time PCRs were carried out on a Step One Plus thermocycler (Applied Biosystems, Thermo Fisher Scientific, Waltham, MA, USA). The standard curve was obtained using Step One™ software and the presence/absence for each sample was analysed with the Software Design and Analysis ver 2.2.1 (Thermo Fisher Scientific, Waltham, MA, USA). Negative (cut-off cycle threshold value Ct > 38), *Neomys a. anomalus*-positive (cut-off value Ct ≤ 38) and dual amplification samples (treated as contaminated) were removed from further analysis (see Table 1). Positive controls for small mammals consisted of a mixture of genetic material from *N. a. anomalus* (DNA from skin) and *G. pyrenaicus* (DNA from intestine).

### 2.4. Hydrolysis Probe Assay, End Point PCR and Design

#### 2.4.1. Real-Time PCR Using Hydrolysis Probes

Detection of microbes and parasites by real-time PCR was performed with commercial kits (Appendix A), except for the detection of *Omphalometra* spp. (see primers and probe in Table 2) and *Listeria* spp. (Appendix A using prs gene following Vitulo [27]). The kits were manufactured by Exopol (Zaragoza, Spain): Kit EXOone *Ascarididae* BASIC; EXOone *Cryptosporidium* spp. BASIC, EXOone *Eimeria* spp. BASIC, EXOone *Salmonella enterica* BASIC, EXOone *C. perfringens* Beta Toxin BASIC, EXOone *Staphylococcus* spp. BASIC and EXOone Pathogenic *Leptospira* BASIC. Real-time PCR assays were performed following the manufacturer’s recommendations and using negative and positive controls (provided). Quantitative single (microbe and parasite identification) and multiplex (mammal species identification) real-time PCR assays were performed in 10 µL reactions containing 4 µL TaqMan ^TM^ Multiplex Master Mix (Applied Biosystems), 1µL of the primer/probe assay and 5 µL of DNA from positive faeces of *G. pyrenaicus* as the template. Our assays contained 1 µM of each oligo and 0.5 µM of the probe. Microbes and parasites controls were provided by manufacturers or from our own collection.

#### 2.4.2. End Point and Real-Time PCR using Hydrolysis Probes to Detect *Omphalometridae*

Further analyses were performed to detect *Omphalometridae* spp. by developing a real time PCR assay in two steps. First, a 1252 bp long partial sequence of the 28S ribosomal DNA gene from *O. flexuosa* (AF300333) [28] was used as the reference sequence to download a collection of the most similar sequences from GenBank. The downloaded dataset included species from the orders Plagiorchiida and Opisthorchiida (Appendix A), which were represented by 82 and 26 genera, respectively, with some additional sequences of unclassified digeneans. These sequences were aligned using MUSCLE, as implemented in MEGA X [29]. This alignment contained 486 sequences and allowed us to locate several potential conserved regions for primer design (Table 2).

One of the primer pairs (Table 2) was selected to amplify a segment of approximately 934 bp (minor size variation among species due to indels) long segment. Additionally, an internal primer was designed at the 3′ end of this fragment to amplify, when necessary, DNA for sequencing, or as a primer for nested PCR (short PCR, Table 2). Secondly, an alignment containing sequences of representatives of *Omphalometra* (AF300333), *Neoglyphe* (AF300329 and AF300330) and *Rubenstrema* (MK585231; AY222275 and AF300331.1) was used to design real-time PCR primers and hydrolysis TaqMan probes to detect these parasites in faeces (Table 2). The newly designed real-time PCR primers targeted a 76 bp fragment of the 28S gene. All faecal samples and total DNA extract from the intestine were screened with this real-time PCR assay. Moreover, one sample of faeces, which tested positive in real-time PCR assay, collected about 20 m from the dead desman was further analysed using microsatellites in order to verify its genetic identity with the dead animal [30].

Total genomic DNA extracted from the intestine of the dead individual was screened using conventional PCR and real-time PCR, as shown in Table 2. Long and short conventional PCRs were performed from each sample in 20 μL reactions containing 0.05 μg template DNA in a master mix prepared with 1 μM of each primer (Table 2), 1× buffer, 1.5 MgCl_2_, nuclease-free water and 0.4 units of BIOTAQ DNA polymerase (BIOLINE, London, UK). PCR protocol was as follows: 5 min at 95 °C, 35 cycles of 60 s at 95 °C, 30 s at 60 °C, 60 s at 72 °C, followed by a final extension step of 10 min at 72 °C. The PCR products were visualized using 1.5% agarose gel electrophoresis and purified with ExSPure (Nimagen, The Netherlands). The long PCR products were sequenced directly using a Big Dye 3.1 cycle sequencing kit (Applied Biosystems, Foster City, CA, USA) on a 3130 Genetic Analyzer (Applied Biosystems, Foster City, CA, USA) using primers used for both the long and short PCRs (Table 3). The resulting sequences were visually inspected in the Sequencing Analysis 5.2 software (Applied Biosystems, Foster City, CA, USA).

Contiguous sequences were assembled using MEGA X (29) and deposited in GenBank (AC N OP407587-OP407588). After trimming primer sequences, the obtained novel 599 bp long sequences were added to the alignment containing 31 sequences of phylogenetically close digenean taxa and *Zalophotrema hepaticum* (GenBank AY222255) as the outgroup; the alignment was trimmed to the length of the shortest sequence. Phylogenetic analyses were conducted using the Likelohood under General Time Reversible nucleotide substitution model as determined by MEGA X. (see Figure 2 for details) with 1000 bootstrap replicates.

### 2.5. Statistical Analysis

A Chi-square analysis by crosstab was performed in IBM SPSS ver 23 software to determine whether there is a significant difference between expected frequencies and the observed frequencies among years, districts, zone and section categories for each molecular target separately, as well as the whole dataset (sum of all positive results within the sample of all detected targets except *Eimeria* spp.). The Fisher’s exact test was carried out using Monte Carlo methods (bilateral, 10^6^ sampling bootstrap and IC 95%) at the *p* ≤ 0.05 level to identify significant differences with Bonferroni corrections. We used *, ** and *** for ≤0.05, ≤0.01 and ≤0.001 significant levels, respectively. Superscripts (a/b) denote subsets where proportions did not differ significantly from each other at the 0.05 level.”

## 3. Results

### 3.1. Sampling Distribution According to the Recuperation Plan of the Desman in Extremadura

Table 3 summarizes the sampling with positive results provided for the administrative local regions as per recuperation plan zoning. The 13.03% of samples yielding water shrews, contaminated or negative real-time PCR were removed from the further analysis. In the Ambroz valley, a critical and favourable area, desmans were well represented, while no samples were obtained at the importance zone despite intensive sampling. Accordingly, the target species was mainly present in all critical zones and, exceptionally, at one favourable zone, Ambroz (Table 3).

### 3.2. Bacteria and Parasites Study

All samples from desmans, including the intestine of the dead desman, were negative for the family Ascarididae. Because the test for the Ascarididae targeted only the genera *Baylisascaris, Parascaris, Ascaris, Toxascaris and Toxocara*, it may be concluded that only species of these genera were absent. Similarly, screening for *Clostridium perfringens* beta toxin was negative and only one sample was positive for *Listeria* spp. The rest of the tests for microbes and parasites found 1.15% to 92.3% positives (Table 3).

The intestine was positive for *Eimeria* spp. and *Omphalometra* spp., which was also confirmed in one faecal sample that was genetically compatible with the dead desman. As a result, the screening for microbes and parasites in desman faeces revealed an overall low prevalence ranging from zero (*Ascaris* or *Clostridium*) to 14.5% for *Staphylococcus* spp. (see Table 3), 1.15% for *Cryptosporidium* spp. and 2.4% for *Leptospira* spp.; the latter two pathogens being a cause of concern due to their pathogenicity and potentially serious impact on the health of the desman [31]. *Eimeria* spp. is excluded due to its surprisingly high prevalence (92.3%) compared with the remaining microbes and parasites, warranting a more in-depth study (in preparation).

After excluding *Eimeria* spp., the combined prevalence of infection by all targeted pathogens was 26.6% (Table 3), with 22.2% of samples containing a single infection, 3.9% containing dual infections and 0.5% having triple infections. The results showed a similar distribution of positives across different years; comments on the differences between administrative district, zoning or sampling section (Table 3) are provided below.

#### 3.2.1. Remarks on Microbes

Screening revealed significant differences in infection prevalence between administrative districts and sampling sections. For instance, *Salmonella* showed a higher prevalence in Ambroz. This fact may be explained by two reasons: (1) five out of six samples were positive in sections 13A and 15A, both located in the favorable area within Ambroz (section; see Table 1), and (2) the desman was only detected in the favourable area in Ambroz, but not in other areas of that district. Instead, *Salmonella* was present in one section within the critical area in La Vera and was absent in Jerte. *Staphylococcus* spp. was less prevalent in La Vera and Jerte compared to Ambroz (Table 3); however, the positives were evenly distributed between the critical and favourable areas in that district. It should be noted, however, that 33.3% of *Staphylococcus*-positive samples were collected in a single critical sampling section (1(A): 10/30; Table 1 and Table 2). Furthermore, infections with pathogenic *Leptospira* spp. were rare.

#### 3.2.2. Remarks on Parasitic Infections

Our study detected a high overall prevalence *of Eimeria* spp., but no significant differences in the prevalence among different sampling years, zones or sampling sections were detected. The same is true for the different administrative districts, except for Ambroz where the prevalence was higher. *Omphalometra* spp. was found in 8.2 % of all samples; it was evenly distributed among years, districts, zones and sampling sections (Table 3), suggesting the availability and broad distribution of the intermediate hosts of these parasites within the habitats of the desman in Extremadura.

The pathogenic *Cryptosporidium* spp. was rare in Ambroz and absent in the remaining districts. However, the low prevalence in Ambroz was due to the occurrence of these pathogens in only two sampling sites, which is relatively good news because of the high pathogenicity of this protozoon.

#### 3.2.3. Prevalence in the Entire Dataset

Statistical analysis of combined data allowed for a general view of the spatial and temporal aspects of the health status of the desman. Only samples from the Ambroz district demonstrated a higher combined prevalence of parasite/bacteria, indicating that the desman populations in the region may be at a greater risk compared to La Vera or Jerte (Table 3). This was less evident in the comparison of the cumulative prevalence between sampling sections, with only Ambroz (9(A) in Table 3) characterized by a higher prevalence of pathogens, especially *Staphylococcus* and *Leptospira* spp.

### 3.3. Ompahlometra Findings

The trematode *O. flexuosa* was commonly reported from the desman [17,18,32,33,34,35]; however, it was not previously recorded in Extremadura. To screen our samples for this parasite, we used a newly developed real-time PCR assay targeting a fragment of the 28S rDNA gene (see Section 2). The results (Table 2) showed a rather even prevalence across years, administrative districts, zones and sampling sections for this parasite. Furthermore, the screening of the dead desman produced a positive result by both real-time and conventional PCR. Furthermore, the real time PCR assay detected *O. flexuosa* both from the faeces found within 20 m from the dead desman on the same collection day and the intestine of the desman itself. In addition, genotyping using nine microsatellite markers adapted from those described by Gillet [30] showed that the faeces and the dead desman were genetically identical.

We were able to obtain sequence of a digenean from the intestine of the dead desman, but were unable to obtain PCR products usable for sequencing from faecal samples. The failure to obtain usable PCR products from faeces was likely due to the low amount of available DNA and/or high-level DNA degradation as suggested by Greiman [36]. A BLAST search of our sequence from the dead desman showed the highest identity to *O. flexuosa* from the snail intermediate host *Planorbis planorbis* in Poland [98.51%] [27] and 95.43% to 95.59% similarity to 28S sequences from *Neoglyphe locellus*, *N. sobolevi* and *Rubenstrema exasperatum*.

Upon the exclusion of six ambiguously aligned nucleotide positions, the contribution of partial 28S sequences used for phylogenetic analysis was 607 bp long. In the phylogenetic tree resulting from the ML analysis (Figure 2), the newly obtained sequences from the desman formed a highly supported (100%) clade with *Omphalometra flexuosa* (AF300333). Based on the results of the phylogenetic analysis and the high pairwise similarity between our newly obtained sequence and *O. flexuosa* (AF300333)*,* we posit that the digenean in our sample belonged to a species of *Omphalometra*. The level of divergence in the 28S sequences suggests that the species in our material is different from the one sequenced from Poland (AF300333). However, until specimens of this digenean become available for detailed morphological and molecular analyses, at this time we opt to call it *Omphalometra* spp.

## 4. Discussion

This is the first study aimed to obtain knowledge about the parasites and pathogens of the most threatened population of the Iberian desman. Unlike previous, mostly opportunistic, reports of helminths in the desman [16,17,18,23,24,27,37], our study covered a broad geographic area in Extremadura and targeted both bacterial and parasitic infections. Although infections by the nematodes *Paracuaria hispanica* and *Aonchotheca galemydis* were reported in the desman, as well as a species of the genus *Maritrema,* known in water shrews from Spain cohabitating with the desman in Extremadura [38,39], these parasites were not studied yet because we could only use commercially available assays for a limited number of nematode taxa. We found a surprisingly high prevalence of *Eimeria* infections, which calls for further research on the subject. However, revealing the true diversity and identity of *Eimeria* spp. in the Iberian desman across its range will require the use of next-generation sequencing approaches. More taxon-specific assays will also need to be developed for the detection and accurate diagnosis of *Paracuaria* and *Aonchotheca,* which are known to parasitize the desman.

### 4.1. The First Overview of the Health Status of the Most Endangered Desman Populations: Central System

This study resulted in the first detection of several pathogens including bacteria, protists and digeneans, some of which have been associated with disease in mammals [10,40,41]. Only six out of eight of the used real-time PCR assays produced positive results by diagnosing six pathogens. Only a single sample tested positive for *Listeria* (0.5%), while the remaining detected parasites and pathogens demonstrated higher prevalence, reaching nearly 15% in *Staphylococcus* (Table 3). Our data suggest that the desman plays a limited role as a reservoir of pathogens that can be shared with other members of mammal community.

This fact may be of importance because many agents of zoonotic diseases in rodents and shrews are often misdiagnosed [10,42]. However, this likely was not the case in our study for several reasons: (1) we used assays to screen for a wide range of microbe and parasite taxa, (2) samples were collected exhaustively to capture the entire spatial distribution of the species in Extremadura and (3) each faecal sample was genetically attributed to the desman. The overall low prevalence of infections and their concentration in discrete areas suggest a limited role of the desman as a host of zoonotic diseases in Extremadura. Nevertheless, despite this, the presence of infections (e.g., *Staphylococcus* spp.) around certain sites within the areas suggests that a special monitoring effort is needed to reveal the sources of infections or whether the circulation of pathogens is maintained within some of these sites, mainly in the Ambroz.

Furthermore, the difference between sites suggested the need for further research to improve our knowledge of suitable habitats for the desman. For example, *Salmonella* could originate from yet unknown reservoirs; the two most likely sources could be the nearest farms with domestic animals and/or contact with other wild mammal species. Most animals are asymptomatic carriers that excrete salmonella intermittently [43], but little is known about the role of wildlife, let alone small mammals, as its reservoirs [44]. Furthermore, possible interactions of rodents and insectivores with farm animals should be taken into account. For example, large-scale poultry or pig farming may be a source of *Salmonella typhimurium* [8], but small-scale or self-sustaining farms typical of the study area may also serve as a source of infection [8]. In the case of *Leptospira* spp., our results show a rather scattered pattern of distribution, which did not allow us to connect it to a possible source. This pathogen is commonly in rodents of the genera *Rattus*, *Mus* and *Apodemus* [45], which often inhabit riverbeds. According to Ospina-Pinto [46], rodents and insectivores commonly harbour the serovars *L. grippotyphosa* and *L. pomona*, both of which were included among the targets of the real-time PCR assays in our study.

Our results on *Staphylococcus* spp. could indicate a significant interaction between the natural habitats and the farms in the surrounding areas. Recent studies have isolated MRSA (methicillin-resistant *Staphylococcus aureus*) in wildlife [47], including small mammals such as *Microtus arvalis*, *Apodemus sylvaticus* and *Rattus norvegicus*, which prompted us to include *Staphylococcus* spp. in this study. However, their clinical significance in many situations is difficult to assess, as they may be either harmless commensals or serious pathogens [48,49]. Consequently, further studies are needed to address this question, as *Staphylococcus* spp. was the most prominent group of bacteria in the desman.

*Eimeria* spp. were of particular interest in our study because they cause coccidiosis in a variety of animals [50]. Although *Eimeria* species were described morphologically from insectivorous mammals [51], molecular data are scarce. Cross-infection studies have shown that coccidia from one host family generally do not infect hosts from other families or orders, although some coccidia are less host specific [51,52,53]. Therefore, considering the high prevalence of *Eimeria* in our study, we plan on untangling the true diversity of *Eimeria* spp. in a separate study. It is known that most coccidia in insectivores are quite specific to their hosts [51]. For example, moles are always infected with multiple intestinal *Eimeria*, while shrews usually harbour only a single species [52]. On the other hand, there is an interest in the screening of rodents and insectivores for *Cryptosporidium* spp., because they can be true reservoirs of *C. parvum* and *C. muris* [54,55]. A high prevalence of *C. parvum*, *C. muris* and *C. tyzzeri* has been reported in small mammals in Spain [54,56], with the former especially widespread in the north-eastern regions [54,57]. These studies have suggested that small mammals, including insectivores, are important sources of transmission of *Cryptosporidium* between wild species even in the absence of farm animals or significant human activity [55]. However, the low prevalence and limited geographic distribution (two sites in Ambroz with only two positives in each site) suggest a very limited intraspecific transmission in desman populations, which is further hindered by the strong isolation between desman populations in Extremadura [1].

### 4.2. New Report of Omphalometra spp. in Desman

Timon David (1960) [16] suggested the desman as the definitive host of the *O. flexuosa* var *peyrei,* which has been considered different from the form parasitizing other talpid species such as *Talpa europaea* [33,35,58]. Its lifecycle is partially known, with snails *Planorbis planorbis* (Linnaeus, 1758) and occasionally *Lymnaea palustris* (Muller, 1774) playing a role as intermediate hosts; both snails are found in Spain [59]. The second intermediate hosts of *Omphalometra* are the larval stages of aquatic insects such as dytiscid diving beetles (Dytiscidae), as well as crustaceans *Gammarus pulex* L., 1758 [17,60]. *Omphalometra flexuosa* var *peyrei* was described as smaller than specimens found in moles [16]. Although the 28s rDNA is a rather conserved gene, our sequence of *Omplalometra* from La Vera was clearly distinguishable from the genera *Neoglyphe* or *Rubenstrema* and was not identical to *O. flexuosa* from moles. Sequences of mtDNA genes could definitively clarify this situation [61]. Unfortunately, mtDNA sequences of *Ompahometra* spp. are currently lacking.

Timon David (1960) [16] dissected 31 *G. pyrenaicus* and found *O. flexuosa* in 2 animals (6.4%), while no other adult helminths were found. Using non-invasive procedures, we detected *Omphalometra* in 8.6% of faecal samples from Extremadura populations, suggesting no major changes in the prevalence of this trematode in the desman. Voucher (1975) also found *O. flexuosa* in desmans from the now extinct population of *G. pyrenaicus* in Candelario (Salamanca, Spain), which is geographically closest to the area surveyed in our study.

In the past, dissection was usually the only way to obtain reliable data on parasites in wildlife, especially small mammals. However, non-invasive strategies followed by real-time PCR, as well as a variety of other techniques, such as NGS or immunodiagnostics, allow us to obtain data about both hosts and parasites without sacrificing or even handling animals [62,63]. The non-invasive approach used in this study allows us not only to understand the current situation regarding pathogen diversity and prevalence in an endangered species such as the desman, but also to identify most suitable habitats for the species persistence, which may help future conservation efforts [64,65,66].

## 5. Conclusions

This study is the first comprehensive survey of pathogens in the most endangered population of *Galemys pyrenaicus* targeting both historically reported and not yet reported microbes and parasites. We can conclude that we found very different prevalences among the etiological agents studied, ranging from none to moderate or high prevalence, such as in the case of *Eimeria* spp., suggesting the need to study all aspects of the biology of these endangered populations, including health. As stated by Figeiredo [14], molecular methods allow for improved diagnostics and more detailed knowledge of the diversity of the microbes and parasitic species of the threatened species themselves, as well as the overlapping communities, to reveal their possible role in the transmission of zoonotic disease agents. Finally, our results indicated that the most endangered population of the desman surveyed in this study requires continuous, multi-faceted monitoring due to the existing and coming threats such as climate change.

## Figures and Tables

**Figure 1 animals-13-01136-f001:**
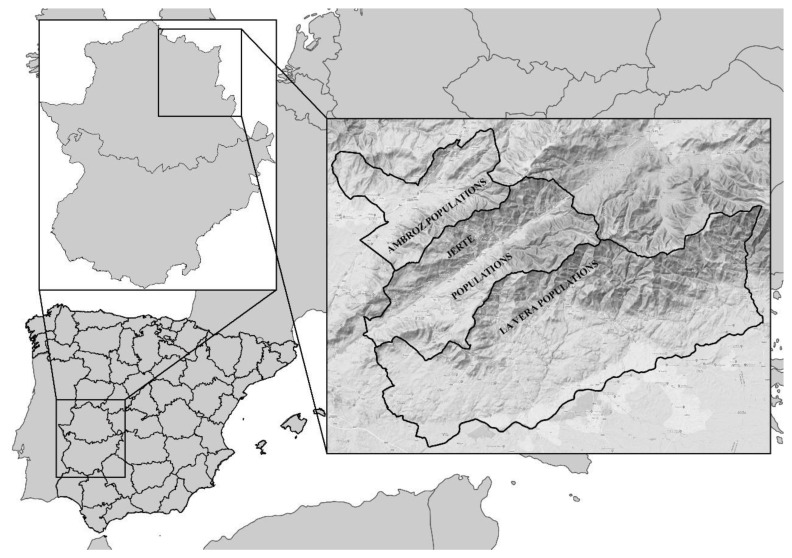
Map locating the three districts with Iberian desman populations in the north of Extremadura.

**Figure 2 animals-13-01136-f002:**
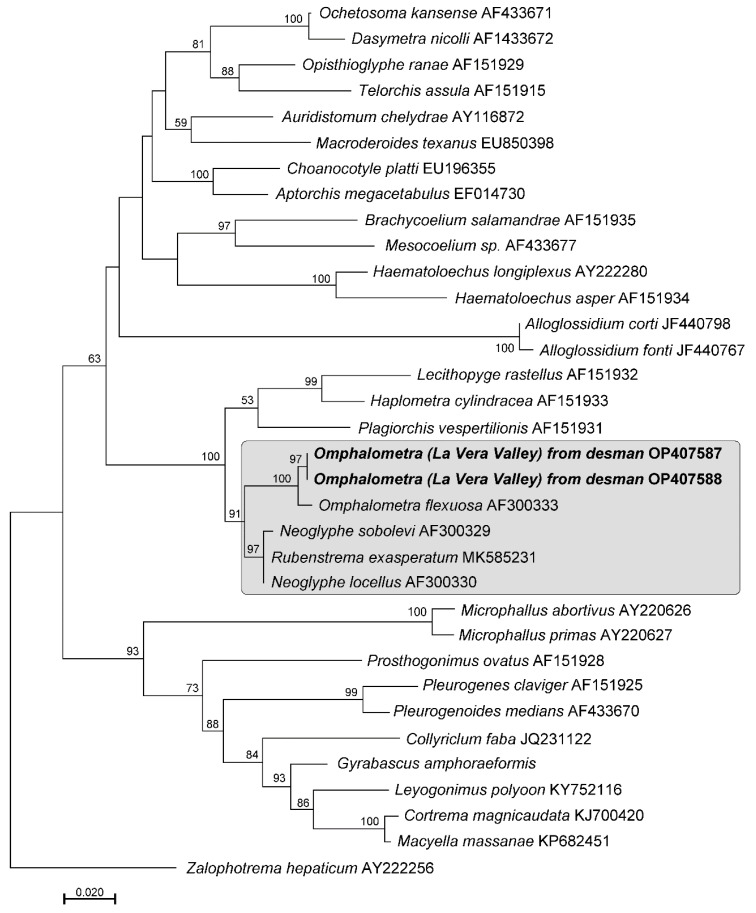
Phylogenetic tree resulting from Maximum Likelihood analysis of 28S rDNA alignment of representative plagiorhioidean and microphalloidean taxa showing the position of *Omphalometra* spp. from the Pyrenean desman sequenced in our study. The family Omphalometridae is indicated by the shaded rectangle. Numbers above internodes indicate bootstrap support values. Only values >50% are shown. The scale bar indicates the number of substitutions per site.

**Table 1 animals-13-01136-t001:** Geographical distribution of samples from 2018 to 2021 by administrative district (A.D.) and sampling sections (S.S.). Percentages are calculated based on all samples (n = 238).

Area Zoning	A.D.	S.S.	S.S. with Desman	Total Samples	Desman	Water Shrews	Both	No Target
				238	207	8	4	19
Critical Zone	Ambroz	n = 4	n = 2	13.45%	10.92%	0.84%	0.00%	1.68%
	Jerte	n = 4	n = 3	30.25%	28.57%	0.00%	0.42%	1.26%
	La Vera	n = 2	n = 2	31.93%	26.89%	0.42%	0.84%	3.78%
			Subtotal	75.63%	66.39%	1.26%	1.26%	6.72%
Importance Zone	Jerte	n = 1	n = 0	0.00%	0.00%	0.00%	0.00%	0.00%
			Subtotal	0.00%	0.00%	0.00%	0.00%	0.00%
Favourable Zone	Ambroz	n = 3	n = 2	22.27%	20.59%	0.42%	0.42%	0.84%
	Jerte	n = 4	n = 0	0.00%	0.00%	0.00%	0.00%	0.00%
	La Vera	n = 6	n = 0	2.10%	0.00%	1.68%	0.00%	0.42%
			Subtotal	24.37%	20.59%	2.10%	0.42%	1.26%

**Table 2 animals-13-01136-t002:** Primers to target *Omphalometridae* and other trematodes.

Size (Gene: 28S rDNA)	Technique	Primer Names	In Silico Species Range
Long PCR: ~934 bp	Conventional PCR	LahpmoF 5′TWCCGBRAGGGAAAGTTGAAA LahpmoR 5′TCACCATCYTTCGGGTCWCA	All in Appendix A.
Short PCR: ~699 bp	Conventional, nested sequencing PCR	LahpmoF 5′TWCCGBRAGGGAAAGTTGAAA Ompahlshort_R 5’TCTCCTTGGTCCGTGTTT	All in Appendix A except *Cercaria nigrospora*
76 bp	Real time PCR	Ompha_F 5′TCAAGTGTGTGCGCTCCG Ompha_R 5′TGCCGGTCGTGGTGACTA Probe 6-FAM-5′TCTCCGGCCTGCTCGTCAGT-BHQ-1	*Omphalometra; Neoglyphe* and *Rubenstrema* in Appendix A.

**Table 3 animals-13-01136-t003:** Prevalence of all infections by year, administrative district, zone and sampling section. Positive assays: C = *Cryptosporidium* spp.; O = *Omphalometra* spp.; E = *Eimeria* spp.; S = *Salmonella* spp.; St = *Staphylococcus* spp.; L = *Leptospira*. (ac = accumulated number of positive assays from 0 to 3) with respect to each sample (*Eimeria* spp. excluded). In bold: *p*-value (95% IC). * and ** for ≤0.05 and ≤0.01 significant level, respectively. Same superscripts (a or b) for proportions did not differ significantly each other Row “+”: Percentage of prevalence of pathogens in desman samples based on real-time PCR screening by target.

	C	O	E	S	St	L	ac (1; 2 or 3)
Year:2019/2020/2021 n = 71/90/46	1/2/0 **0.797 (0.789–0.805)**	5/8/4 **0.899 (0.893–0.905)**	63/83/45 **0.201 (0.193–0.209)**	4/1/4 **0.073 (0.068–0.078)**	9/18/3 **0.095 (0.089–0.101)**	1/2/2 **0.720 (0.711–0.729)**	16;1;1/19;6;0/ 11;1;0 **0.591 (0.582–0.601)**
Admin. District (A vs. J vs. V) n = 75/68/64	3/0/0 **0.109 (0.103–0.115)**	5/8/4 **0.489 (0.479–0.498)**	67/67/57 **0.042 * (0.038–0.046)**	6/0/3 **0.043 * (0.039–0.047)**	19 ^a^/6 ^b^/5 ^b^ **0.006 ** (0.005–0.008)**	2/3/0 **0.321 (0.312–0.331)**	24;6;0 ^a^/ 12;1;1 ^b^/ 10;1;0 ^b^ **0.013 ** (0.011–0.053)**
Zone (Crit. vs. Fav.) n = 158/49	1/2 **0.140**	15/2 **0.371**	149/42 **0.650**	4/5 **0.036 *****	22/8 **0.649**	5/0 **0.594**	31;7;1/15;1;0 **0.426 (0.416–0.436)**
Sampling sections							
8 (A) n = 30	2	2	26	1	4 ^a,b^	0	7/1/0 ^a,b^
9 (A) n =19	1	3	19	0	10 ^a^	2	6/5/0 ^a^
13 (A) n= 7	0	0	6	1	1 ^a,b^	0	3/0/0 ^a,b^
14 (A) n= 19	0	0	16	4	4 ^a,b^	0	8/0/0 ^a,b^
16 (J) n = 22	0	2	22	0	2 ^a,b^	0	4/0/0 ^a,b^
21 (J) n = 7	0	2	7	0	1 ^a,b^	1	1/0/1 ^a,b^
24 (J) n = 39	0	4	38	0	3 ^b^	2	7/1/0 ^a,b^
10 (V) n= 31	0	3	28	0	3 ^b^	0	6/0/0 ^a,b^
19 (V) n = 33	0	1	29	3	2 ^b^	0	4/1/0 ^b^
	**0.256 (0.248–0.265)**	**0.335 (0.326–0.344)**	**0.194 (0.186–0.202)**	**0.007 ** (0.005–0.008)**	**0.004 ** (0.002–0.005)**	**0.063 (0.059–0.068)**	**0.008 ** (0.006–0.009)**
**+**	**1.15%**	**8.2%**	**92.3%**	**4.3%**	**14.5%**	**2.4%**	**26.6%**

## Data Availability

The data presented in this study are available on request from the author.

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
