# Peer review of "Non-Invasive Wildlife Disease Surveillance Using Real Time PCR Assays: The Case of the Endangered Galemys pyrenaicus Populations from the Central System Mountains (Extremadura, Spain)"

_animals, 2023, doi:10.3390/ani13071136_

Round 1
Reviewer 1 Report
The authors present some parasites and bacteria found in fecal samples of Galemys pyrenaicus (desman) in an area of Spain.
The study is well-designed, and the results are new and interesting. Nevertheless, I want to suggest some changes to the manuscript. The M&M section should be rewritten and reorganized as it is difficult to view all the procedures thoroughly.
Material and methods:
Table 1 states that samples were obtained from desman and water shrews. Were the samples from the shrews analyzed for pathogens? If not, I recommended eliminating this information from the table as it is confusing.
Table 2 shows only results, therefore should be moved to the result section. What is the accumulated number of positive assays? What represents *, superindex a and b?
2.2 DNA extraction also describes the identification of the samples. Please split it
Table S1-S3 and S5-S7: have the authors checked all the organisms stated in those tables, or is it just a copy of the manufactures instructions? If it is the second, I suggest keeping the commercially available kit’s name and removing the published information.
Table 3, I think, is a preliminary version. Please update the line for real-time PCR
2.3 When describing the PCR conditions, please add concentration and no volume so it can be reproduced somewhere else
Line 173: misspelling problem: “inteste” should be “intestine”
It is difficult to follow all PCR and tests carried out in the different samples: shrews, desman, and intestine, as section 2.3 explains the PCRs performed, and the one developed all mixed. I recommend fragmenting this section or adding a diagram that helps the reader to follow all procedures.
Results,
Line 214 is repetitive with line 212
Table 4 should be added to table 2.
In lines 226-234, italics are missing
Line 231-232: misspelling problem: seriou simpact -- serious impact
Discussion:
The authors state that Paracuaria hispanica, Aonchotheca galemydis, and Maritrema were undetected. I recommend changing line 318 to “these parasites were not studied” instead of not detected as it can lead to confusion and misinterpretation
Line 344: please rephrase as it is not clear the idea
Lines 301 and 387 are contradictory
Conclusions:
Line 419: lacks italics
Author Response
Response to Reviewer 1 Comments
Point 1: The authors present some parasites and bacteria found in fecal samples of Galemys pyrenaicus (desman) in an area of Spain. The study is well-designed, and the results are new and interesting. Nevertheless, I want to suggest some changes to the manuscript. The M&M section should be rewritten and reorganized as it is difficult to view all the procedures thoroughly.
Material and methods
Table 1 states that samples were obtained from desman and water shrews. Were the samples from the shrews analysed for pathogens? If not, I recommended eliminating this information from the table as it is confusing.
Response 1: Samples from shrews were not further analysed because they did not belong to desmans: "All faecal samples were genotyped to ensure that they belonged to desman".
Our mistake was in line 158 where we wrote "(see Results)" instead of "(see Table 1)". We corrected the error.
Point 2: Table 2 shows only results, therefore should be moved to the result section.
Response 2: We thank the reviewer for spotting our error. We have corrected some mistakes in tables 1 and 2. Now, this table 2 change to table 3.
Point 3: What is the accumulated number of positive assays?Table 2 shows only results
Response 3: Please see Table 3 and 4 but now rejoin as Table 3.
Point 4:. What represents *, superindex a and b?
Response 4: We have added the following information: “The Fisher's exact test was carried out using Monte Carlo methods (bilateral, 106 sampling bootstrap and IC 95%) to identify significant differences at p ≤ .05 level with Bonferroni corrections. *, ** and *** for ≤.05, ≤.01 and ≤.001 significant levels, respectively. Superscripts (a/b) denote subsets where proportions did not differ significantly from each other at the .05 level.”
Point 5:. 2.2 DNA extraction also describes the identification of the samples. Please split it
Response 5: Done as suggested. I have split that section and re-numbered the sections in the Materials and methods.
Point 6: Table S1-S3 and S5-S7: have the authors checked all the organisms stated in those tables, or is it just a copy of the manufactures instructions? If it is the second, I suggest keeping the commercially available kit’s name and removing the published information.
Response 6: In the case of the commercial KIT, the information on all detectable organisms is not in the manufacturer's instructions but is only reported as xxxx. spp. so it is not a copy of the KIT. So, it was the manufacturer who provided us with this information at our request (personal communication). They suggested that this information could appear in the study. Therefore, we would like to keep the list sent by the manufacturer and thank him for his courtesy in the article as we informed to manufacturers, unless otherwise indicated.
Point 7: Table 3, I think, is a preliminary version. Please update the line for real-time PCR
Reponse 7: Thank you. We have completed the table.
Point 8: 2.3 When describing the PCR conditions, please add concentration and no volume so it can be reproduced somewhere else
Reponse 8: We agree that it would be better, but in this case the manufacturer of the kit (EXOone) does not provide concentrations, only volumes.
Our assays used 1microM of each oligo and 0.5 microM of the probe. It was corrected.
Point 9: Line 173: misspelling problem: “inteste” should be “intestine”.
Reponse 9: Corrected.
Point 10: It is difficult to follow all PCR and tests carried out in the different samples: shrews, desman, and intestine, as section 2.3 explains the PCRs performed, and the one developed all mixed. I recommend fragmenting this section or adding a diagram that helps the reader to follow all procedures
Reponse 10: As explained above and in the text, upon initial genotyping no further analyses were performed on shrew samples. Only faeces and one intestine from desman were analysed.
We divided the 2.3 section.
Point 11: Line 214 is repetitive with line 212.
Reponse 11: Corrected.
Point 12: Table 4 should be added to table 2.
Reponse 12: Done as suggested.
Point 13: In lines 226-234, italics are missing.
Reponse 13: Corrected.
Point 14: Line 231-232: misspelling problem: seriou simpact -- serious impact.
Reponse 14: Corrected.
Point 15: The authors state that Paracuaria hispanica, Aonchotheca galemydis, and Maritrema were undetected. I recommend changing line 318 to “these parasites were not studied” instead of not detected as it can lead to confusion and misinterpretation.
Reponse 15: Changed as suggested.
Point 16: Line 344: please rephrase as it is not clear the idea.
Reponse 16: Re-phrased as " Nevertheless, despite this, the presence of infections (e.g. Staphylococcus spp.) around certain sites within the areas suggests that a special monitoring effort is needed to reveal the sources of infections or whether the circulation of pathogens is maintained within some of these sites, mainly in the Ambroz."
Point 17: Lines 301 and 387 are contradictory.
Reponse 17: Thanks, you for spotting this error. Corrected.
Point 18: Line 419: lacks italics.
Reponse 18: Corrected.

Reviewer 2 Report
The main remark I would like to make concerns the "bacteriological" part of the study. In my opinion, this part of the work has not been done as well as parasite studies, and raises a number of questions. It is not clear to me why the authors chose this particular set of pathogenic bacteria for detection. Why does the set of bacterial pathogens include precisely these bacterial species, and not include more epidemically significant pathogens, and/or pathogens more specific for wild mammals (at least F. tularensis)? Why did the authors not detect viral infections at all? Having a DNA/RNA preparation isolated from a field sample, the authors could use this preparation to detect a much wider list of pathogens. This would not greatly increase the cost of the work, but the results would be much more informative.
I realize that the work has already been done, and it is unlikely that now there is an opportunity to expand the list of detected infections. But in my opinion, the authors should somewhat rewrite the Introduction section and indicate the relevance for desmans of precisely those pathogens that the avors detected, or somehow justify their choice of a set of detectable pathogens.
Minor remarks:
1) Table 2.
It is not indicated by what methods the authors determined the infection of animals. Is this the authors' own results? If yes, then this data should be transferred to the "results" section.
Also there is no interpretation of the indices "a" and "b" in the legend of this table .
2) lines 361-367
It seems to me that the reasoning of the authors about MRSA is somewhat unfounded here. The authors did not show the presence of MRSA in the studied samples using PCR or cultural methods. In the current version of the manuscript, the data are presented in such a way that it is not obvious to the reader whether Staphylococcus spp is the normoflora of the desman, or whether the presence of these bacteria in the test sample indicates a disease in the animal.
3) Tables S1 - S7.
Too many tables listing the same type of data. Wouldn't it be better to combine them into one table?
Table S8, "Accession Number" column.
It is worth specifying more clearly which Accesion Number the authors have in mind, that is, where exactly to look for the specified sequence.
Author Response
Response to Reviewer 2 Comments
Point 1: The main remark I would like to make concerns the "bacteriological" part of the study.
In my opinion, this part of the work has not been done as well as parasite studies, and raises a number of questions.
It is not clear to me why the authors chose this particular set of pathogenic bacteria for detection.
Why does the set of bacterial pathogens include precisely these bacterial species, and not include more epidemically significant pathogens, and/or pathogens more specific for wild mammals (at least F. tularensis)?
Response 1: We agree with their statement. Although the study of bacteria and protists is not comprehensive, it is a significant step forward because of the lack of previous data on desman. The taxonomic coverage of available diagnostic assays is limited to major pathogens known in small mammals. Moreover, some of the tests were developed specifically by us or requested from the manufacturing company.
Due to the state (degradation as the result of exposure to elements) and size (droppings are less than 200 milligrams in most cases) of the faecal samples, they were limited for analyses. The resources available for the study were not endless, therefore we could not apply every potentially available method of analysis at this point. We now know the habitats of the desman in Extremadura and hope to be able to carry out more in-depth studies of its microbiota in the future. Some of them are currently in the early stages of research in our Laboratory.
Point 2: Why did the authors not detect viral infections at all? Having a DNA/RNA preparation isolated from a field sample, the authors could use this preparation to detect a much wider list of pathogens. This would not greatly increase the cost of the work, but the results would be much more informative.
Response 2: Of course, we think about the viruses too, but not everything can be done in one study. We first focused on helminths, protist and some bacteria and plan to study viral infections at a later point.
Point 3: I realize that the work has already been done, and it is unlikely that now there is an opportunity to expand the list of detected infections. But in my opinion, the authors should somewhat rewrite the Introduction section and indicate the relevance for desmans of precisely those pathogens that the avors detected, or somehow justify their choice of a set of detectable pathogens.
Response 3: We agree. We added the following text:
“Due to the state (degradation due to exposure to elements) and size (droppings are less than 200 milligrams in most cases) we focused on the groups of parasites known from desman (such as digeneans) or important pathogens reported from other small mammals, for which we could develop our own assays or obtain commercially available diagnostic kits. Based on previously available knowledge from insectivores or other small mammals, the pathogens covered in our study, are among those commonly affecting animal health."
Point 4:. 1) Table 2. It is not indicated by what methods the authors determined the infection of animals. Is this the authors' own results? If yes, then this data should be transferred to the "results" section.
Response 4: It has been transferred to the results section. The information can be found in the Table 3 due to some changes. in the order in the revised text.
Point 5:. Also there is no interpretation of the indices "a" and "b" in the legend of this table
Response 5:. We apologize, the explanation was missing in the submitted version of the manuscript. Then we added:
“Same superscript (a/b) denotes subsets where proportions did not differ significantly from each other at the .05 level.”
Point 6:. 2) lines 361-367. It seems to me that the reasoning of the authors about MRSA is somewhat unfounded here. The authors did not show the presence of MRSA in the studied samples using PCR or cultural methods. In the current version of the manuscript, the data are presented in such a way that it is not obvious to the reader whether Staphylococcus spp is the normoflora of the desman, or whether the presence of these bacteria in the test sample indicates a disease in the animal
Response 6: The reviewer is correct. We did not intend to study MRSA at this stage. However, our first intention was to look for the presence of Staphylococcus spp., as some articles seem to suggest. However, these articles focused mainly on MRSA Staphylococcus in wildlife (which is widespread among micromammals). This encouraged us to investigate it, but first we needed to obtain evidence for the presence of Staphylococcus spp. (wide range of species) in studied sites, and the location of their sources. We now know where to locate the sources of this bacterium in the populations, which will be useful for further studies to screen for MRSA more specifically. We have also rewritten the paragraph to make it easier for the reader to understand:
"Our results on Staphylococcus spp. could indicate a significant interaction between the natural habitats and the farms in the surrounding areas. Recent studies have isolated MRSA (methicillin-resistant Staphylococcus aureus) in wildlife [47], including small mammals such as Microtus arvalis, Apodemus sylvaticus and Rattus norvegicus, which prompted us to include Staphylococcus spp. in this study. However, their clinical significance in many situations is difficult to assess, as they may be either harmless commensals or serious pathogens [48,49]. Consequently, further studies are needed to address this question, as Staphylococcus spp. was the most prominent group of bacterium in desman."
Point 7: 3) Tables S1 - S7.Too many tables listing the same type of data. Wouldn't it be better to combine them into one table?
Reponse 7: The supplementary tables were reduced to S1 and S2.
Point 8: Table S8, "Accession Number" column. It is worth specifying more clearly which Accesion Number the authors have in mind, that is, where exactly to look for the specified sequence.
Reponse 8: Corrected as suggested.
